# Insulin-Like Growth Factor-Binding Protein 7 (IGFBP-7)—New Diagnostic and Prognostic Marker in Symptomatic Peripheral Arterial Disease?—Pilot Study

**DOI:** 10.3390/biom12050712

**Published:** 2022-05-17

**Authors:** Anna Szyszkowska, Sylwia Barańska, Robert Sawicki, Ewa Tarasiuk, Marlena Dubatówka, Marcin Kondraciuk, Emilia Sawicka-Śmiarowska, Małgorzata Knapp, Jerzy Głowiński, Karol Kamiński, Anna Lisowska

**Affiliations:** 1Department of Cardiology, Medical University of Bialystok, 15-276 Bialystok, Poland; annaszyszkowska92@gmail.com (A.S.); r-sawicki@o2.pl (R.S.); ewa-tarasiuk@o2.pl (E.T.); emiliasawickak@gmail.com (E.S.-Ś.); malgo33@interia.pl (M.K.); fizklin@wp.pl (K.K.); 2Department of Vascular Surgery and Transplantation, Medical University of Bialystok, 15-276 Bialystok, Poland; lebie_1f@op.pl (S.B.); jglow@wp.pl (J.G.); 3Department of Population Medicine and Lifestyle Diseases Prevention, Medical University of Bialystok, 15-276 Bialystok, Poland; m.paniczko@gmail.com (M.D.); marcin.kondraciuk@umb.edu.pl (M.K.)

**Keywords:** IGFBP-7, peripheral artery disease, atherosclerosis

## Abstract

The aim of our study was to evaluate the importance of insulin-like growth-factor-binding protein 7 (IGFBP-7) as a potential marker of symptomatic peripheral artery disease (PAD) occurrence. The study group consisted of 145 patients with diagnosed PAD, who qualified for the invasive treatment. The control group consisted of 67 individuals representing the local population and an ischemic heart disease (IHD) group of 88 patients after myocardial infarction or percutaneous coronary intervention. Patients with PAD had significantly higher IGFBP-7 concentrations than control group (1.80 ± 1.62 vs. 1.41 ± 0.45 ng/mL, *p* = 0.04). No significant differences between PAD patients and IHD patients were found (1.80 ± 1.62 vs. 1.76 ± 1.04 ng/mL, *p* = 0.783). Patients with multilevel PAD presented significantly higher IGFBP-7 concentrations than patients with aortoiliac PAD—median 1.18 (IQR 0.48–2.23) vs. 1.42 ng/mL (0.71–2.63), *p* = 0.035. In the group of patients who died or had a major adverse cardiovascular event (MACE) during six months of follow-up, a statistically significant higher IGFBP-7 concentration was found (median 2.66 (IQR 1.80–4.93) vs. 1.36 ng/mL (IQR 0.65–2.34), *p* = 0.004). It seems that IGFBP-7 is elevated in patients with atherosclerotic lesions—regardless of their locations. Further research should be conducted to verify IGFBP-7 usefulness as a predictor of MACE or death.

## 1. Introduction

Atherosclerotic disease of the lower extremities, also known as peripheral arterial disease (PAD), is a significant and increasing clinical problem. It is estimated that more than 200 million people worldwide have PAD [1,2]. The symptoms vary from intermittent claudication (IC), atypical leg pain, critical limb ischemia, and acute limb ischemia [3], which is a state of a sudden decrease in limb perfusion, caused by acute arterial occlusion of a pre-existing stenotic arterial segment (60% of cases) or a result of an embolic event, dissection of an artery, or a direct artery trauma, in a previously asymptomatic patient [4].

PAD can be asymptomatic, which is several times more common than IC [1]. Both asymptomatic and symptomatic PAD is linked to a significantly increased risk of cardiovascular morbidity and mortality [3]. A total of 20% of patients with IC within 5 years will have a major cardiovascular event (stroke or myocardial infarction, MI) and 10–15% of them will die—75% due to cardiovascular reasons. The most common cause of death (40–60%) is ischemic heart disease (IHD), 10–20% of deaths are caused by cerebral artery disease, and 10% by other vascular events, such as a ruptured aortic aneurysm. For patients with acute ischemia, short-term mortality is 15–20% [5]. Moreover, through walking impairment and often the need for amputation, PAD is associated with significant disability worldwide [1].

Common cardiovascular risk factors, such as age [1,2,3,5,6,7,8], gender [1,2,3,5,8], smoking [1,2,3,5,6,7,8], diabetes [1,2,3,5,6,7,8], hypertension [1,2,3,5,6,7,8], and hypercholesterolemia [1,2,3,5,6,7,8] contribute to the development and progression of PAD. Considering the aging of the population and global increases in smoking and diabetes, it will be a major public health challenge in the future [3]. Despite these undeniable facts, PAD still receives relatively little research and public attention. Therefore, there is a need to study these issues, such as searching for specific biomarkers that will have diagnostic and prognostic value in this group of patients. One of them may be the insulin-like growth factor binding protein 7 (IGFBP7)—a novel marker of cellular senescence, insulin resistance, and atherosclerosis [9,10]. IGFBP-7, as a part of the insulin-like growth factor system, is involved in the growth, proliferation, and differentiation of human cells. According to the latest research, it could play a significant role in the highly complex pathogenesis of atherosclerosis [9,10]. Significantly higher levels of IGFBP-7 were observed in the group of patients with stable IHD and after MI than in the control group of healthy subjects [11]. The importance of IGFBP-7 in the development of PAD has not been studied so far—there are no data on this topic in the literature. Our study is the first one handling this issue. Considering the fact that patients with PAD are at particularly high cardiovascular risk and require early diagnosis and intensive treatment to prevent complications and death, searching for new biomarkers in this group of patients is crucial.

The aim of our study was to evaluate the importance of IGFBP-7 as a potential marker of symptomatic PAD occurrence, to evaluate the correlation of IGFBP-7 concentration with classical cardiovascular risk factors in this group of patients, and to assess the usefulness of IGFBP-7 as a prognostic marker after lower limbs arteries revascularization.

## 2. Materials and Methods

### 2.1. Study Population

One hundred and forty-five patients with diagnosed PAD, who qualified for the invasive (surgical or endovascular) treatment and were admitted to the Department of Vascular Surgery and Transplantation between 2018 and 2021, were prospectively evaluated. The indications for invasive treatment were assessed in the Vascular Surgery Outpatient Clinic and consisted of IC distance significantly impeding daily functioning (individualized for each patient depending on the overall clinical picture, but no more than 200 m—despite marching training and optimal pharmacotherapy), presence of rest pains, and the threat of limb amputation.

In all studies, patients’ baseline characteristics regarding demographics, smoking status, medical history, concomitant diseases, diagnostic test results, and pharmacology, were collected. Moreover, intermittent claudication distance was obtained, and every patient was assessed with the Fontaine scale. The only exclusion criterion was the lack of permission from the patient.

### 2.2. Control Group

The control group consisted of 67 individuals, recruited from participants of the population-based cohort study Bialystok PLUS [10,12], which represents the general local population in age 20–80. Moreover, we compare the study group to patients with ischemic heart disease (IHD). The IHD group consisted of 88 patients with established IHD approximately a year after myocardial infarction or percutaneous coronary intervention, recruited to one of the cardiology centers taking part in the multinational registry EUROASPIRE V [13]. The general characteristics of the control and IHD groups are presented in Table 1. A more detailed description of the control group and IHD group was published previously [10].

### 2.3. Ethical Issues

The study was approved by the Local Bioethics Committee Nr R-I-002/39/2019. Informed consent was obtained from all subjects involved in the study.

### 2.4. Biochemical Evaluation

The studied material was venous blood drawn during the first 24 h after the patient’s admission to the hospital, before surgical intervention. The patients were fasting during blood collection. In order to obtain analyzed biochemical parameters (total cholesterol, low-density lipoprotein (LDL) cholesterol; high-density lipoprotein (HDL) cholesterol; triglycerides, glucose, creatinine, and IGFBP-7 concentration), the blood was drawn to clot in a closed system of the Monovette type (SARSTEDT, Nümbrecht, Germany). Biochemical parameters were determined within 2 h after the material’s collection. The blood samples (5 mL) drawn for IGFBP-7 determination were left at room temperature for 2 h to allow clot formation and then centrifuged at 1000× *g* for 20 min at room temperature. Freshly prepared supernatant serum was frozen and stored at −80 °C until use. The concentration of IGFBP-7 was established with commercially available ELISA kit 7 (insulin growth-factor-binding protein; USCN Life Science Inc., Wuhan, Hubei, China), according to the manufacturer’s instructions. The obtained results are the mean of two almost identical measurements. In the control group, blood samples were collected during outpatient care visits. Then, IGFBP-7 levels were compared between PAD patients and control groups. The glomerular filtration rate was calculated by the Cockroft–Gault formula. In all participants, blood pressure and anthropometric measurements (height, weight) were performed. Body mass index (BMI) was calculated as standard. Blood pressure (BP) was measured using the oscillometric method after the participants were seated for at least 10 min. Ankle–brachial index (ABI) was measured using the ABPI MD equipment (MESI, Ljubljana, Slovenia).

### 2.5. Statistical Analysis

The mean values and standard deviations for quantitative variables as well as the quantitative and percentage distribution for qualitative variables were calculated. Data were presented as means (%) and standard deviation (SD) distributed continuous variables, medians (Me) and interquartile range (IQR) for not normally distributed continuous variables, and as the number (N) of cases and percentage (%) for categorical variable. Statistical significance of differences between the two groups was determined using the *t*-test (for comparing normal continuous variables) and the Mann–Whitney U test (for comparing non-normal continuous variables). Pearson’s correlation coefficient for categorical variables of normal distribution and Spearman’s correlation coefficient for variables not satisfying normal distribution criteria were calculated. The comparison of qualitative variables between the groups was performed using the Chi2 test. To compare multiple groups NIR test was used. For univariate analysis, Odds Ratio Calculator was used. The statistical analysis was carried out using Statistica 12.0 PL software. A value of *p* < 0.05 was considered statistically significant.

## 3. Results

### 3.1. Characteristics of the Study Group

The study population was characterized by very high cardiovascular risk. A total of 93.1% of patients smoked at least in their past with the average number of pack-years 38.4 ± 19.8. A total of 80.7% of them had hypertension, 69.7% dyslipidemia, 42.8% had type 2 diabetes, and 30.3% had IHD. According to the Fontaine scale, 67.6% of patients belonged to stage IIb, and 31.0% to stage III. In two patients, the Fontaine scale assessment was unavailable due to the lack of one limb (amputation in medical history). A total of 129 patients (88.9%) had an IC distance below 100 m. What is very alarming, 105 patients (91%) had LDL cholesterol level above 55 mg/dL (mean 94.5 ± 36.5 mg/dL).

The study group consisted of 109 men (75.2%) and 36 women (24.8%). Women qualified for invasive treatment were in worse conditions than men. They had a shorter distance of IC—median 20 m (IQR 10–50) vs. 50 m (IQR 20–100), *p* = 0.02, and 50% of them were in stage III according to the Fontaine scale (in comparison to this stage was 24.8% of men, *p* = 0.004). They also had diabetes more often than men (61.1% vs. 36.7%, *p* = 0.009). The detailed characteristics of the study group with gender comparison are presented in Table 2.

Treatment during the hospitalization and after the discharge from the hospital is presented in Table 3.

The eldest patients in the study group (>75 years, *n* = 38) had significantly higher concentrations of IGFBP-7 than patients < 60 years and patients with age between 61–75 years—median 1.89 ng/mL (IQR 1.01–2.78) vs. 1.04 ng/mL (IQR 0.55–2.08, *p* = 0.026) and vs. 1.33 ng/mL (IQR 0.47–2.56, *p* = 0.04), respectively. Moreover, they presented lower hemoglobin levels—12.1 ± 2.1 mg/dL vs. 14.1 ± 1.4 mg/dL (*p* < 0.001) and vs. 13.3 ± 1.9 mg/dL (*p* = 0.002) and worse kidney function—eGFR 66.6 ± 26.6 mL/min/1.73 m^2^ vs. 103.4 ± 28.3 mL/min/1.73 m^2^ (*p* < 0.001) and vs. 81.5 ± 33.4 mL/min/1.73 m^2^ (*p* = 0.02), respectively. They also had shorter IC distance, compared to the youngest patients—median 20 m (IQR 10–50) vs. 50 m (IQR 20–100), *p* = 0.027. The detailed comparison of the age groups is presented in Table 4.

Based on the extension of the atherosclerotic lesions described in the computed tomography (CT) scans, the patients were divided into four groups: patients with aortoiliac occlusive disease (16p. (11.0%)), femoropopliteal occlusive disease (35p. (24.1%)), tibial occlusive disease (5p. (3.5%)) and multilevel peripheral artery disease (89p. (61.4%)). Patients with tibial occlusive disease, due to the small size of the group, have been omitted from the calculations. Patients with atherosclerotic lesions (≥50% stenosis in the angio-CT scan) in at least two areas of the limbs or after invasive treatment in an area other than the one operated during current hospitalization, were qualified as patients with multilevel PAD. Compared to the subjects with other types of PAD, they were older (67.2 ± 8.0 years vs. 69.2 ± 8.8 years, *p* = 0.2), had worse kidney function (creatinine concentration 1.3 ± 1.5 mg/dL vs. 1.02 ± 0.4 mg/dL, *p* = 0.2), and lower hemoglobin levels (12.8 ± 2.0 mg/dL vs. 13.6 ± 1.8 mg/dL, *p* = 0.01)—but only hemoglobin concentrations were statistically significant. There were no statistically significant differences between the incidents of comorbidities in patients with multilevel PAD, compared to subjects with other types of PAD.

The patients underwent five types of invasive procedures: balloon angioplasty without stent implantation (40p. (27.6%)), balloon angioplasty with stent implantation (70p. (48.3%)), reconstructive surgery with a bypass graft from the patient’s vein (2p. (1.4%)), reconstructive surgery with an artificial bypass graft (10p. (6.9%)) and hybrid procedures (15p. (10.3%)). For the statistical calculations, reconstructive surgery with a bypass graft from the patient’s vein and reconstructive surgery with an artificial bypass graft were combined. Eight patients were disqualified from the invasive PAD treatment, due to severe kidney failure (1p.), recent MI (1p.), arrhythmia (1p.), anesthetic disqualification (1p.), and extensive purulent lesions (1p.), long IC (3p.).

### 3.2. IGFBP-7 Measurements

Patients with PAD had a significantly higher IGFBP-7 concentration than the control group—1.80 ± 1.62 ng/mL vs. 1.41 ± 0.45 ng/mL, *p* = 0.04. No significant differences between PAD patients and IHD patients were found—1.80 ± 1.62 ng/mL vs. 1.76 ± 1.04 ng/mL, *p* = 0.78—Figure 1.

Patients with multilevel peripheral artery disease presented significantly higher IGFBP-7 concentrations than patients with aortoiliac occlusive disease—median 1.18 ng/mL (IQR 0.48–2.23) vs. 1.42 ng/mL (0.71–2.63), *p* = 0.035. There were no statistical differences in the rest group of patients—Figure 2.

Patients who were disqualified from invasive procedures had higher IGFBP-7 concentrations than those who qualified for the surgery, but without statistical significance—median 2.84 ng/mL (IQR 0.86–5.88) vs. 1.40 ng/mL (IQR 0.65–2.46), *p* = 0.09.

We have also found that diabetic patients treated with insulin had significantly higher IGFBP-7 concentrations than those treated with oral hypoglycemic drugs—median 0.86 ng/mL (IQR 0.19—1.95) vs. 1.57 ng/mL (IQR 1.12–4.5), *p* = 0.027, respectively—Figure 3.

Significant, but weak positive correlations of IGFBP-7 and age (r = 0.23, *p* = 0.004), creatinine (r = 0.39, *p* < 0.001) and negative correlations with eGFR (r = −0.40, *p* < 0.001) and hemoglobin (r = −0.18, *p* = 0.03) in the study group were found.

### 3.3. Follow-Up

Six months after hospitalization in the Department of Vascular Surgery and Transplantation, a telephone visit was performed. Thirty-four (23.45%) patients had another invasive procedure on the lower limb arteries and thirteen patients (8.97%) had an endpoint during the follow-up: three (2.01%) had a stroke, six (4.14%) had a heart attack and six (4.14%) died. Two patients died due to MI, one patient due to Coronavirus Disease 2019 (COVID19) infection, and in three patients the cause of death was unclear. Eleven patients were lost to follow-up. Reasons for another invasive procedure on the lower limb arteries were: significant stenosis in another area of the lower limb arteries than the operated one (assessed in the angio-CT scan), and recurrence of clinical symptoms. There were no statistically significant differences between patients who underwent another invasive procedure for PAD and the rest of the group.

Patients who had a major cardiovascular event (MI or stroke) or who died, had worse kidney function (serum creatinine 2.11 ± 2.05 vs. 1.05 ± 0.59 mg/dL, *p* < 0.001, eGFR 51.18 ± 28.99 vs. 82.7 ± 31.43 mL/min/1.73 m^2^), lower hemoglobin levels (11.26 ± 1.79 vs. 13.28 ± 1.89 mg/dL, *p* < 0.001) and more often were diagnosed with diabetes (72.73% vs. 39.34%, *p* = 0.03). Moreover, they had statistically higher IGFBP-7 concentrations than the survivors—median 2.66 ng/mL (IQR 1.80–4.93) vs. 1.36 ng/mL (IQR 0.65–2.34), *p* = 0.004.

The univariate analysis confirmed the significance of the following factors influencing the appearance of the major cardiovascular event (MI or stroke) and death: diabetes (*p* = 0.02, odds ratio [OR] = 4.11, 95%CI 1.039–16.272), total cholesterol > 125 mg/dL (*p* = 0.0002, OR = 11.25, 95%CI 2.912–43.456), HDL cholesterol ≤ 40 mg/dL (*p* = 0.01, OR = 6.16, 95%CI 1.271–29.88), eGFR ≤ 55 mL/min/1.73 m^2^ (*p* = 0.002, OR = 6.79, 95%CI 1.842–25.033), hemoglobin ≤ 12.6 mg/dL (*p* = 0.003, OR = 18.37, 95%CI 2.275–148.376), and IGFBP-7 concentration > 2.40 ng/mL (*p* = 0.001, OR = 8.55, 95%CI 2.128–34.363).

Multivariate analyses were not performed due to the low power of statistical analysis resulting from the small number of endpoints (MI, stroke, or death).

## 4. Discussion

IGFBP-7 is proven to be involved in the development of many diseases, including diabetes, obesity, acute kidney injury (AKI), and cancers. According to our previous research, it seems that it could play an important role in the development of atherosclerosis [10,11]. However, little is known about its role in PAD. Our research, which is as far as we know the first one about the importance of IGFBP-7 in PAD development, seems to confirm the possible role of this protein in peripheral arteries atherosclerosis.

Due to the lack of data in the literature on the role of IGFBP-7 in PAD, we may only refer to the studies carried out for serum IGFBP2 levels in this group of patients. Taking into account the fact that IGFBP2 belongs to the same group of proteins that combines with insulin and insulin-like growth factors (IGFs), this kind of analysis appears appropriate. Researchers discovered that an increased IGFBP2 concentration was significantly and independently associated with long-term cardiovascular disease mortality in patients with PAD, but it should not be regarded as a useful clinical marker in CVD (cardiovascular disease) mortality prediction in this population of patients [14]. In our research, we focused on IGFBP-7. It has a 500 times higher affinity to insulin than other IGFBPs and it is proven to be associated with insulin resistance (IR), development of metabolic syndrome (MetS), and consequently cardiovascular diseases [9,15,16].

According to our findings, patients with PAD have significantly higher concentrations of IGFBP-7 than individuals representing the local population. Those with multilevel peripheral artery disease had a significantly higher concentration than patients with the aortoiliac occlusive disease; however, due to the small size of the study group and highly asymmetrical distribution, this data could be inaccurate and may depend on confounding factors. There were no significant differences between other types of PAD.

This data is consistent with our previous research, in which we observed that IGFBP-7 concentrations were significantly higher in patients with coronary artery atherosclerosis compared to the control group of healthy individuals [10,11]. However, IGFBP-7 levels did not differ significantly between MI and stable IHD patients [11]. There were no significant relationships between IGFBP-7 concentrations and the extent of coronary lesions [11].

Patients who were disqualified from invasive procedures had higher IGFBP-7 concentrations than those who qualified for the surgery, but without statistical significance, which may be a result of the small size of the disqualified group. Most of the disqualified patients were in a poor condition—which may be the reason for the IGFBP-7 higher concentrations; however, none of them died during the 6-months follow-up.

Moreover, according to our research, IGFBP-7 concentration did not significantly differ between patients with PAD and patients with IHD. It could mean that IGFBP-7 is elevated in patients with atherosclerotic lesions—regardless of their locations. Our previous study, in which we discovered a significant correlation between IGFBP-7 and intima-media thickness in carotid arteries [11], may also support this hypothesis.

All subjects in the control, IHD, and PAD groups had additional cardiovascular risk factors, but only patients with confirmed arteriosclerosis of the lower extremities or coronary arteries had significantly elevated IGFBP-7 levels. It may suggest IGFBP-7 correlation with atherosclerotic lesions. Further research on the accurate dependence of IGFBP-7 and atherosclerosis should be conducted.

As mentioned earlier, IGFBP-7 is proven to be associated with insulin resistance (IR) and the development of metabolic syndrome (MetS) [9,15,16]. It competes with insulin receptors (InsR) for binding insulin, which causes a reduction in the serum level of free insulin, blocks insulin binding with InsR, and diminishes the physiological response to insulin. As a result, it could contribute to insulin resistance and MetS, which significantly increases the risk of developing diabetes and IHD [9,15,16] According to research, patients with MetS have a five-fold greater risk of developing DM and a two-fold greater risk for IHD in comparison to those without MetS [17]. In research on the Chinese population, patients with MetS and IR had significantly higher serum IGFBP-7 levels than control healthy subjects [15]. Taking into account the IGFBP-7 role in insulin metabolism, it may be a new potential target for the treatment of IR and MetS [9,15]. In our research, we found that diabetic patients treated with insulin had significantly higher IGFBP-7 concentrations than those treated with oral drugs, which is consistent with our previous research [10] and seems to correspond with findings on the role of this protein in insulin resistance and diabetes development.

According to our findings, IGFBP-7 significantly correlates with serum parameters of kidney function (creatinine, eGFR), which is consistent with our previous studies [9]. Other researchers have also demonstrated that IGFBP-7 together with tissue inhibitor of metalloproteinases-2 (TIMP-2) are good markers of tubular damage [9,18,19]. Urinary TIMP2*IGFBP and IGFBP-7 were shown to be the most accurate biomarker of prediction and renal outcome in patients with AKI [9,10,19].

We have also discovered that patients who died or had MACE (MI or stroke) during 6 months of follow-up, had statistically higher IGFBP-7 concentrations, than the survivors. However, due to the small size of the group, this data should be considered with caution. To confirm these reports, a larger cohort of patients and long-term follow-up should be conducted.

In our study group, 24.8% of patients were women, who had a shorter distance of IC than men, and more often they suffered from rest pain (III stage in the Fontaine scale). According to data, worldwide women accounted for 23–52% of people with peripheral artery disease [3]. The prevalence of PAD in high-income countries seems to be similar in men and women. However, in low-income countries, it seems to be higher for women [7]. Interestingly, IC incidence in women is about half that in men [1,20,21], but they more often suffer from atypical pain [1]. In general, the more severe or symptomatic disease is more common in men [1,8]—it could be the reason for late PAD diagnosis in females, which is consistent with our research.

In our study group, most of the patients had multi-level PAD, and very few had tibial PAD, which, considering the fact that 42.8% of them were diabetics, may be surprising. However, some of the patients with multi-level PAD had atherosclerotic lesions in tibial arteries together with stenosis in another area of the limbs. It may also be due to the fact that isolated tibial PAD is less frequently qualified for invasive treatment than other types of PAD. The arteries below the knee are small and tend to be calcified. Moreover, the atherosclerotic lesions in this area are often diffuse. According to the literature, due to the high complexity of these changes in the tibial area, invasive treatment should be limited to patients with chronic limb-threatening ischemia [22].

According to the European Society of Cardiology (ESC) Guidelines on cardiovascular disease prevention in clinical practice [23], published in 2021, subjects with PAD are at very-high cardiovascular risk and lipid-lowering treatment with an ultimate LDL-C goal of <55 mg/dL is recommended. In our study, the mean value of LDL-C was 101.9 ± 44.8 mg/dL and 91% of patients had LDL cholesterol levels above 55 mg/dL. This is very alarming data, and it shows that more focus on efficient lipid-lowering treatment should be put on.

The study limitation of our research is the small size of the study group; however, this is the first study handling the issue of the possible IGFBP-7 role in PAD development. Further research on this topic should be conducted. Moreover, the IGFBP-7 cut points have not been prospectively validated yet and probably vary from population to population. Finally, we did not assess ABI in PAD patients due to logistic reasons (every patient had an angio-CT scan of the lower limbs arteries prior to the hospital admission).

However, our pilot study is the first one handling the issue of the IGFBP-7 role in PAD patients. To our knowledge, no such studies have been conducted in this population and, as mentioned earlier, these are patients at particularly high cardiovascular risk, who require early diagnosis and intensive treatment to prevent complications and death.

## 5. Conclusions

Patients with PAD have significantly higher concentrations of IGFBP-7 than individuals recruited from the general population, but there were no statistically significant differences between PAD and IHD patients. It could mean that IGFBP-7 is elevated in patients with atherosclerotic lesions—regardless of their locations. Moreover, patients who died or had MACE (MI or stroke) during 6 months of follow-up, had statistically higher IGFBP-7 concentrations than the survivors, although to confirm these reports, a larger cohort of patients and long-term follow-up would be needed. Further research on the role of IGFBP-7 in atherosclerosis and PAD development is required.

## Figures and Tables

**Figure 1 biomolecules-12-00712-f001:**
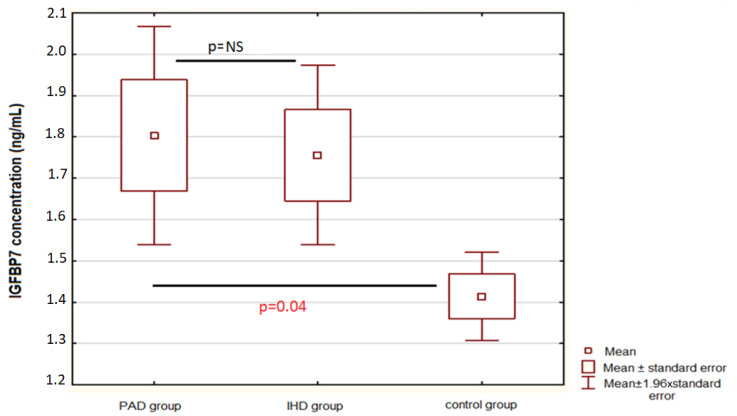
Comparisons between patients with PAD, IHD, and the control group according to the IGFBP-7 concentration. Abbreviations: IGFBP7—insulin-like growth-factor-binding protein 7, IHD—ischemic heart disease, PAD—peripheral artery disease.

**Figure 2 biomolecules-12-00712-f002:**
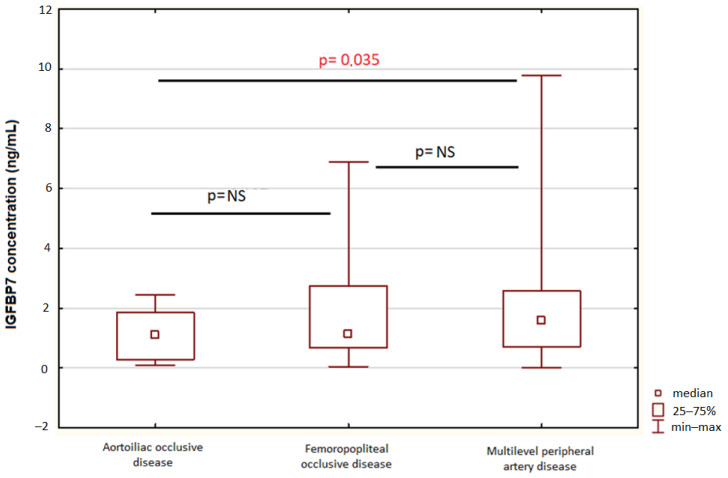
IGFBP-7 concentrations in patients with aortoiliac, femoropopliteal, and multilevel PAD. Abbreviations: IGFBP-7—insulin-like growth-factor-binding protein 7.

**Figure 3 biomolecules-12-00712-f003:**
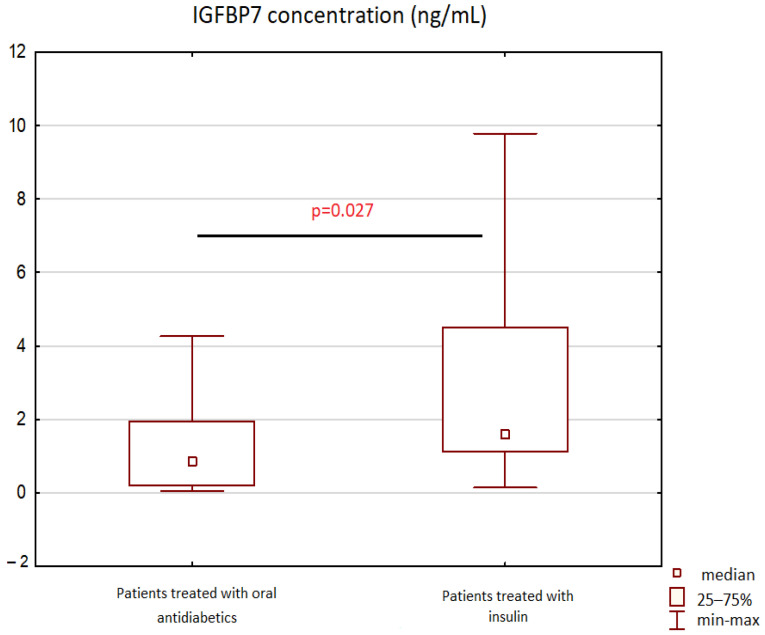
IGFBP-7 concentration in patients on insulin therapy vs. patients treated with oral antidiabetics. Abbreviations: IGFBP-7—insulin-like growth-factor-binding protein 7.

**Table 1 biomolecules-12-00712-t001:** The general characteristics of the control and IHD groups.

	Control Group (*n* = 67)	IHD Group (*n* = 88)
Age (years)	61.2 ± 8.2	63.2 ± 7.8
Male gender, n (%)	41p. (61.2%)	69p. (78.4%)
Ankle-brachial index	1.2 ± 0.09	1.2 ± 0.12
Hypertension, n (%)	48p. (78.6%)	79p. (89.8%)
Diabetes t.2, n (%)	40p. (59.7%)	69p. (78.4%)
Ischemic heart disease, n (%)	0p. (0%)	88p. (100%)
Hiperlipidemia, n (%)	56p. (83.6%)	80p. (90.9%)
Peripheral artery disease, n (%)	0p. (0%)	4p. (4.5%)

Abbreviations: IHD—ischemic heart disease.

**Table 2 biomolecules-12-00712-t002:** Patient characteristics with gender comparison.

	Study Group (*n* = 145)	Men (*n* = 109, 75.2%)	Women (*n* = 36, 24.8%)	*p*
Age, years	68.8 ± 8.2	68.1 ± 8.1	71.0 ± 8.3	0.07
BMI, kg/m^2^	27.3 ± 4.9	26.9 ± 4.3	28.4 ± 6.1	0.12
Smoking in the present, n (%)	60p. (41.4%)	44p. (40.4%)	16p. (43.8%)	0.45
Smoking in the past, n (%)	135p. (93.1%)	104p. (95.4%)	31p. (86.1%)	0.08
Number of pack-years, years	38.4 ± 19.8	38.0 ± 21.7	23.1 ± 19.0	0.01
Intermittent claudication distance, metres, median (IQR)	50.0 (15.0–100.0)	50.0 (20.0–100.0)	20.0 (10.0–50.0)	0.02
Stage IIb according to Fontaine scale, n (%)	98p. (67.6%)	81p. (74.3%)	17p. (47.2%)	0.004
Stage III according to Fontaine scale, n (%)	45p. (31.0%)	27p. (24.8%)	18p. (50.0%)	0.004
Systolic BP, mmHg	136.6 ± 13.6	135.8 ± 13.2	139.2 ± 14.6	0.19
Diastolic BP, mmHg	78.6 ± 9.1	79.2 ± 9.2	76.9 ± 8.9	0.19
Hypertension, n (%)	117p. (80.7%)	83 (76.2%)	34 (94.4%)	0.01
Hiperlipidemia, n (%)	101p. (69.7%)	77p. (70.6%)	24p. (66.7%)	0.4
Diabetes t.2, n (%)	62p. (42.8%)	40p. (36.7%)	22p. (61.1%)	0.009
Ischemic heart disease, n (%)	44p. (30.3%)	34p. (31.2%)	10p. (27.8%)	0.44
Total cholesterol, mg/dL	165.0 ± 45.8	168.1 ± 49.4	154.7 ± 30.2	0.16
LDL-cholesterol, mg/dL	94.5 ± 36.5	97.0 ± 39.2	86.1 ± 24.4	0.17
HDL-cholesterol, mg/dL	44.6 ± 15.0	43.6 ± 14.7	47.6 ± 15.7	0.18
TG, mg/dL	135.1 ± 88.1	137.2 ± 91.3	128.6 ± 78.3	0.65
Glucose, mg/dL	131.6 ± 55.2	129.1 ± 51.8	139.8 ± 65.4	0.36
Creatinine, md/dL	1.1 ± 0.8	1.1 ± 0.9	1.2 ± 0.6	0.67
eGFR, ml/min/1.73 m^2^	81.2 ± 32.9	86.4 ± 32.8	65.5 ± 28.5	0.001
Hemoglobin, mg/dL	13.1 ± 2.0	13.5 ± 1.9	12.0 ± 1.8	0.0001
IGFBP-7 concentration, ng/mL, median (IQR)	1.42 (0.67–2.54)	1.62 (1.18–2.83)	1.15 (0.56–2.53)	0.1
Previous invasive treatment of PAD, n (%)	60p. (41.4%)	43p. (39.5%)	17p. (47.2%)	0.27
Aortoiliac occlusive disease, n (%)	16p. (11.0%)	15p. (13.8%)	1p. (2.8%)	0.06
Femoropopliteal occlusive disease, n (%)	35p. (24.1%)	27p. (24.8%)	8p. (22.2%)	0.47
Tibial occlusive disease, n (%)	5p. (3.5%)	3p. (2.8%)	2p. (5.6%)	0.36
Multilevel peripheral artery disease, n (%)	89p. (61.4%)	64p. (58.7%)	25p. (69.4%)	0.17
Ballon angioplasty without stent, n (%)	40p.(27.6%)	28p. (25.7%)	12p. (33.3%)	0.25
Ballon angioplasty with stent implantation, n (%)	70p. (48.3%)	55p. (50.5%)	15p. (41.7%)	0.24
Reconstructive surgery with a bypass graft from the patient’s vein, n (%)	2p. (1.4%)	1p. (0.9%)	1p. (2.8%)	0.44
Reconstructive surgery with an artificial bypass graft, n (%)	10p. (6.9%)	6p. (5.5%)	4p. (11.1%)	0.31
Hybrid procedures, n (%)	15p. (10.3%)	13p. (11.9%)	2p. (5.6%)	0.23
Disqualification from an invasive procedure, n (%)	8p. (5.5%)	6p. (5.5%)	2p. (5.6%)	0.64

Abbreviations: BMI—Body Mass Index; BP—blood pressure; eGFR—Estimated Glomerular Filtration Rate; HDL—high-density lipoprotein; IGFBP-7—insulin-like growth-factor-binding protein 7; IHD—ischemic heart disease; LDL—low-density lipoprotein; TG—triglycerides.

**Table 3 biomolecules-12-00712-t003:** Treatment during the hospitalization and after the discharge from the hospital.

Drug	Number of Patients%
Aspirin	130p. (89.7)
Clopidogrel	110p. (75.9)
Oral anticoagulant	30p. (20.7)
Cilostazol	25p. (17.2)
ACE inhibitors	57p. (39.3)
ARBs	40p. (27.6)
Beta-blockers	85p. (58.6)
Statins	135p. (93.1)
Oral antidiabetic	37p. (25.5)
Insulin	33p. (22.8)

**Table 4 biomolecules-12-00712-t004:** Comparison between age groups.

	Group 1 Age < 60 (*n* = 24)	Group 2 Age 61–75 (*n* = 83)	Group 3 Age > 75 (*n* = 38)	Group 1 vs. Group 3 (*p*)	Group 2 vs. Group 3 (*p*)
BMI, kg/m^2^	28.5 ± 5.9	26.8 ± 4.5	27.6 ± 4.8	0.5	0.37
Intermittent claudication distance, metres, median (IQR)	50.0 (10.0–100.0)	50.0 (20.0–100.0)	20.0 (10.0–50.0)	0.39	0.027
Stage IIb according to Fontaine scale, n (%)	17p. (70.8%)	62p. (75.6%)	19p. (51.4%)	0.11	0.009
Stage III according to Fontaine scale, n (%)	7p. (29.2%)	20p. (24.4%)	18p. (48.7%)	0.11	0.009
Hypertension, n (%)	17p. (70.8%)	66p. (79.5%)	34p. (91.9%)	0.036	0.07
Hiperlipidemia, n (%)	18p. (75.0%)	53p. (63.9%)	30p. (79.0%)	0.48	0.07
Diabetes t.2, n (%)	8p. (33.3%)	40p. (48.2%)	14p. (36.8%)	0.5	0.17
Ischemic heart disease, n (%)	7p. (29.2%)	23p. (27.7%)	14p. (36.8%)	0.37	0.21
Total cholesterol, mg/dL	181.0 ± 59.4	163.8 ± 43.6	156.7 ± 38.4	0.07	0.43
LDL-cholesterol, mg/dL	101.9 ± 44.8	93.0 ± 35.8	93.8 ± 33.6	0.49	0.92
HDL-cholesterol, mg/dL	44.7 ± 16.8	45.2 ± 14.7	43.2 ± 14.9	0.73	0.52
TG, mg/dL	169.6 ± 158.0	134.7 ± 68.0	112.6 ± 52.6	0.07	0.11
Glucose, mg/dL	125.9 ± 52.8	138.8 ± 60.1	118.4 ± 41.8	0.58	0.08
Creatinine, md/dL	0.9 ± 0.2	1.1 ± 0.9	1.3 ± 1.0	0.027	0.31
eGFR, ml/min/1.73 m^2^	103.4 ± 28.3	81.5 ± 33.4	66.6 ± 26.6	<0.0001	0.02
Hemoglobin, mg/dL	14.1 ± 1.4	13.3 ± 1.9	12.1 ± 2.1	<0.0001	0.002
IGFBP-7 concentration, ng/mL, median (IQR)	1.0 (0.6–2.1)	1.3 (0.5–2.6)	1.9 (1.0–2.8)	0.026	0.04

Abbreviations: BMI—Body Mass Index; BP—blood pressure; eGFR—Estimated Glomerular Filtration Rate; HDL—high-density lipoprotein; IGFBP-7—insulin-like growth-factor-binding protein 7; IHD—ischemic heart disease; LDL—low-density lipoprotein; TG—triglycerides.

## Data Availability

Data may be available on reasonable scientific request to the senior author.

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
