# Peer review of "Insulin-Like Growth Factor-Binding Protein 7 (IGFBP-7)—New Diagnostic and Prognostic Marker in Symptomatic Peripheral Arterial Disease?—Pilot Study"

_biomolecules, 2022, doi:10.3390/biom12050712_

Round 1

Reviewer 1 Report

It was a pleasure reviewing this study by Szyszkowska et al, and I congratulate the authors on all their hard work. Please find my comments for different sections of the manuscript below:

Introduction

  1. The authors need to distinguish acute limb ischemia from PAD, as not every patient with acute limb ischemia has PAD. For instance, a patient may have acute ischemic leg secondary to an embolus from the heart due to atrial fibrillation. Therefore, there needs to be a distinction between acute limb ischemia and acute on chronic limb ischemia due to PAD [PAD is a chronic disease]. (line 45-47)

Methodology

  1. The inclusion criteria needs to be further clarified. It appears that only patients requiring a surgical intervention [open or endovascular] were recruited. However, what about stable PAD patients – were they recruited? If only patients requiring revascularization were recruited, then the data obtained cannot be generalized to the PAD patient population. As I am sure the authors are aware, patients who require arterial revascularization is a small subset of the PAD population and therefore the findings from this study are not generalization to the PAD patient population as a whole. The authors may need to re-write the paper focusing on the utility of this biomarker within this specific patient population (i.e. PAD patients requiring surgical intervention).
  2. What were the indications that necessitated the invasive treatment? Was it patients with chronic limb threatening ischemia? Or did the authors operate on patients with claudication? This needs to be clarified
  3. Can the authors present the ankle-brachial index values for patients within the control group?
  4. Is there contaminant PAD in the IHD group? Most patients with IHD have PAD, and thus may be a potential confounding factor. An ABI measurement in the IHD group is crucial here
  5. Can the authors clarify if the patients were fasting while withdrawal? This is important especially in the measurement of lipids.

Results

  1. The number of patients in table 1 do not appear to be adding up. For instance, there appears to be 62 patients with diabetes within the entire study, but there also 77 males with diabetes? Similarly, there are 101 patients with hyperlipidemia overall but there are only 34 males and 10 females with hyperlipidemia. The authors need to revise and review this table to ensure accuracy.
  2. It appears that the patient population recruited to the study is biased to patients with moderate to severe PAD. If so, the data cannot be generalizable to the entire PAD population which compromises of over 60% asymptomatic’ s as well as patients with mild and moderate disease. This is reflected in the demographics within the table such percentage of LDL, DM, smoking etc. Thus, the selection of patients is biased and does not represent the general PAD population which is problematic.
  3. Can the authors comment on why very few of the patients appear to be on an oral hypoglycemic/ insulin while over 70% of the patients are diabetic? Based on table 2, it becomes apparent that the admitted patients who are post-intervention are not managed with best medical therapy – highlighting a major issue.
  4. In lines 160-168, please add the missing units eg walking distance, age etc.
  5. Is IGFBP7 renally cleared? This is an important confounding factor which needs to be assessed and needs to be mentioned in the discussions section.
  6. I find the patients with tibial disease to be extremely low, which is highly unusual in a patient population like the one recruited in this study. Based on what we see in clinical practice, with the 77 male and 24 female diabetics, there should be over 5 patients with tibial disease. We are aware that CT is not the best modality to assess tibial disease due to medial artery calcification- especially in a patient population where diabetes is prevalent such as in the study. The authors need to comment on this atypical finding
  7. The units of the IGFBP7 are missing in some sections of the paper – please add.
  8. The authors state that the levels of IGFBP7 are lower in the control group that the PAD group, however, I would like to see the demographics for the control and IHD. The reason for this as the observed findings could be due to confounding resulting from cardiovascular risk factors (e.g hypertension, hyperlipidemia etc.), which would call into question the observed findings of elevated IGFBP7 due to atherosclerosis/PAD. The authors need to account for such potential confounding factors and should comment on the fact that these factors may play a major role in releasing IGFBP7 into circulation. The improper defining of the control group as the lack of statistical analysis challenges the interpretation of the observed findings.
  9. Figure 2 - higher levels were observed in patients with multi – vessel disease, which could be attributed to PAD, but these patients are typically “unwell” and do have other uncontrolled factors such as hypertension, DM etc. Again, these confounding factors need to be addressed before a justifiable claim can be made by the authors.
  10. The confidence intervals in figure 2 are very wide which stems from the low sample size as well as the non-normal distribution of the patient populations. This raises concerns about the effect of other confounding factors in these patients.
  11. How was the multi-vessel disease defined? Was a specific stenosis in a vascular bed needed? For example, at least >50 in tibial + at least >50 in aorto-iliac??
  12. How many patients were lost to follow-up? Please add this information.
  13. What were the indications for their intervention in patients? Were they re-occurrence of clinical symptoms versus stenosis observed on radiographical assessment? Please kindly clarify.
  14. The authors conducted statistical analysis on patients who passed away, keeping in mind the low sample size (n=6). Without an adequate sample size, I don’t think the authors can conduct any statistical analysis, due to underpowering. The same holds true for the univariate analysis, where the sample size is extremely low and as such, the findings do not yield any certainty. Furthermore, the confidence interval for the calculated odds ratio is extremely wide, reflecting the non-normal distribution as well as low sample size [odds ratio 4.11, 95%CI 039-16.272, OR=11.25, 95%CI 2.912-43.456, CI 1.842-25.033, CI 2.275-148.376]. Such analysis could not be performed and should not be included within this trial.

Discussion

  1. In the start of the discussion section, the authors indicated potential confounding factors that may to increase the level of IGFBP7 (obesity, diabetes, renal disease). However, the authors did not make apparent the demographics for the control group within the study. Therefore the observed findings could truly be due to the presence of the confounding factors and not due to the presence of PAD. This threatens the main conclusion of this trial and poses as a major observed weakness. This is supported in the observed findings as well since the authors did not observe any difference between patients with IHD and PAD. This indicates that the common cardiovascular risk factors among both diseases contribute to IGFBP7 elevation in this patient population and not necessary the presence of PAD. The authors need to address this within the discussion section.
  2. The authors claimed that a passed away patient also had elevated IGFBP7 levels. Clinically speaking, I suspect that these individuals are “sicker” due to the presence of confounding factors, which may have resulted in their demise. This observation may not have anything to do with PAD, based on the information provided thus far by the authors in the manuscript.

Author Response

We appreciate very much the reviewers for their interest in our work and for helpful, constructive comments that will greatly improve the manuscript. We have tried to do our best to respond to the points raised and we have made necessary changes according to the reviewers’ indications. Please see the attachment. 

Reviewer 2 Report

The authors tried to respond to a question:  Insulin-like growth factor-binding protein 7 (IGFBP7) – new diagnostic and prognostic marker in peripheral arterial disease? Please see my suggestions regarding this manuscript:

The appearance of the work is very sloppy. Remove the empty spaces between the paragraphs. Bellow you will find also more suggestions related to the aspect of the manuscript.

L36, 165, 231, 242, 245, 250, etc. Please correct the numerical value (with point, not with comma - English style). Revise the entire manuscript in this regard.

L50. Each reference from 1 to 7 must be inserted after each cardiovascular risk factors. 

L63-67. Please complete the aim of the study. Which is the novelty this paper brings to the field, or at least - the special aspects? There are numerous papers in the field. What makes your paper special?

Table 2. Second cell in the head of the table should be 

Number of patients (%) 

and remove the symbol % near each numerical value in the 2nd column.

Figures titles. Title of each figure should be under the Figure, not above it. Please check the Instructions for authors.

Table 3, L242, 250, etc. is about m2 not m2. Please correct in the entire manuscript.

On Figures 1 and 2 the numerical values must be written in English style. 

The Discussion chapter needs to be improved/completed. Please provide data in a summarised (like a table) form from pervious studies where IGFB7 was associated with atherosclerotic lesions. Please describe potential therapeutical ways to influence the activity of IGFB7, could the new antidiabetic drugs influence its activity ((I suggest checking Vesa C.M., Current Data Regarding the Relationship Between Type 2 Diabetes Mellitus and Cardiovascular Risk Factors, Diagnostics 2020, 10(5), 314. https://doi.org/10.3390/diagnostics10050314). Please discuss whether IGFB7 levels can be influenced by insulin-resistance or not? (maybe finding useful Bungau S. et al. Interactions between leptin and insulin resistance in patients with pre diabetes, with and without NAFLD, Exp. Ther. Med. 2020, 20(6), 197. https://doi.org/10.3892/etm.2020.9327) Could you explain why insulin tretaed patients had higher levels of IGFB7 levels? Is it cost efficient to determine this biomarker for atherosclerosis risk calculation?

L331-336. You have added only the limitations. Please add also the strengths of your research.

Author Response

(The authors gave the same response as above.)

Round 2

Reviewer 1 Report

I would like to congratulate the authors on their work. This was a very interesting manuscript and it's always exciting to see the advancement in PAD research.

Reviewer 2 Report

The authors responded to my suggestions.